# An observational study to examine how cumulative impact zones influence alcohol availability from different types of licensed outlets in an inner London Borough

Carolyn A Sharpe,[1,2] Alan Poots,[3] Hilary Watt,[1] Chris Williamson,[4] David Franklin,[5] Richard J Pinder[1,2]

[1]Department of Primary Care and Public Health, School of Public Health, Imperial College London, London, UK
[2]Public Health Directorate, Southwark Council, London, UK
[3]NIHR CLARHC Northwest London, Imperial College London, London, UK
[4]Public Health Division, Place and Wellbeing Department, Southwark Council, London, UK
[5]Licensing, Southwark Council, London, UK

**Correspondence to**
Carolyn A Sharpe;
carolyn.sharpe15@imperial.ac.uk

## ABSTRACT

**Objectives** Cumulative impact zones (CIZs) are a widely implemented local policy intended to restrict alcohol availability in areas proliferated with licensed outlets. Limited previous research has questioned their effectiveness and suggested they may play a more nuanced role in shaping local alcohol environments. This study evaluates the association between CIZ implementation and the number of licence applications made, and the number issued, relative to a control region.

**Design** A quantitative observational study.

**Setting** The inner London Borough of Southwark, which currently enforces three CIZs.

**Population** Licence applications received by Southwark Council's Licensing Authority between 1 April 2006 and 31 March 2017 (n=1254).

**Interventions** CIZ implementation.

**Primary outcome measures** Five outlet types were categorised and evaluated: drinking establishments, eateries, takeaways, off sales and other outlets. Primary outcome measures were the number of applications received and the number of licences issued. These were analysed using Poisson regression of counts over time.

**Results** Across all CIZs, implementation was associated with greater increases in the number of eateries in CIZ regions (incidence rate ratio (IRR)=1.58, 95% CI: 1.02–2.45, p=0.04) and number of takeaway venues (IRR=3.89, 95% CI: 1.32–11.49, p=0.01), relative to the control area. No discernible association was found for the remaining outlet types. Disaggregating by area indicated a 10-fold relative increase in the number of new eateries in Peckham CIZ (IRR=10.38, 95% CI: 1.39–77.66, p=0.02) and a fourfold relative increase in the number of newly licensed takeaways in Bankside CIZ (IRR=4.38, 95% CI: 1.20–15.91, p=0.03).

**Conclusions** CIZs may be useful as policy levers to shape local alcohol environments to support the licensing goals of specific geographical areas and diversify the night-time economy.

## Strengths and limitations of this study

► This study provides the first quantitative evaluation of cumulative impact zones that disaggregates outlet type into five discrete categories.
► This research provides deeper insight into the nature of alcohol exposure in different locations within an urban centre and therefore provides an evidence base from which local policy can be developed to support local licensing goals which may include diversification of the night-time economy.
► This study shows that descriptive analyses of alcohol control policies (using absolute numbers) are insufficient, and provides a robust statistical approach that can be employed in other urban settings.
► Although substantial steps were taken to assure the reproducibility of the results, the categorisation of outlet type into five discrete categories involved qualitative assessment and there remains a risk of subjectivity.
► Necessary stratification of the dataset rendered some strata with very few (or in some cases zero) counts, thereby constraining statistical power.

## INTRODUCTION

### Alcohol harm and opportunities to control availability

Alcohol occupies a prominent role in modern society. It is a commodity used by many, across social strata, as a relaxant and a means of enjoyment.[1] The misuse of alcohol leads to increased morbidity and mortality. In England in 2016, it was estimated that almost 13 million adults drank at levels that increased their risk of harm.[2] Among those aged 15–49, alcohol has become the leading risk factor for ill-health, early mortality and disability.[3]

Outlet spatial density is positively associated with consumption.[4 5] Policies to restrict the physical availability of alcohol are well

supported in the literature.[6] However, such evidence is based on aggregate studies that consider all alcohol-outlet types as equal. At best, outlets have been categorised as either on-sales or off-sales. On-sales of alcohol refer to venues such as pubs, nightclubs and restaurants where alcohol is purchased to be consumed on the premises. Off-sales of alcohol refer to premises such as off-licences and supermarkets from which alcohol is purchased to be consumed away from the retailer's premises.[7] Although greater association has been found between off-sales and alcohol-related hospitalisations and deaths than with on-sales,[8] both on-licences and off-licences cover a broad range of establishments. Such aggregate studies remain a crude way to evaluate alcohol availability[9] and debate remains around when to use aggregated and disaggregated analyses.[10]

A full description of the licensing process in England and Wales, as defined by the Licensing Act 2003,[11] is outlined in our previous paper.[12] In summary, revised guidance issued under section 182 of the Licensing Act 2003 enables local licensing authorities (LAs) to address the cumulative impact of a concentration of licensed premises.[13] Cumulative impact is defined as 'the potential impact on the promotion of the licensing objectives of a significant number of licensed premises concentrated in one area'.[13] The licensing objectives, in England and Wales, as defined by the Licensing Act 2003 are: the prevention of crime and disorder, public safety, the prevention of public nuisance and the protection of children from harm.[11]

A cumulative impact zone (CIZ) may therefore be established, if a LA can evidentially justify that the licensing objectives are not being upheld in a particular location. This evidence must be published in a LA's statement of licensing policy (SOLP).[13] The Licensing Act 2003 requires all LAs to develop and publish a SOLP at least every 5 years.[11] Evidence to establish a CIZ relates to the licensing objectives. Local crime, disorder and antisocial behaviour statistics can be evaluated alongside acute health data, such as alcohol-related ambulance attendances and hospital admissions. Environmental health complaints, particularly in relation to litter and noise, can also be included.[13]

### Current evidence relating to CIZs
In 2016 there were 215 CIZs established across 106 LAs.[14] Despite their widespread establishment, there is a scarcity of literature evaluating their impact.[15] Revised guidance issued under section 182 of the Licensing Act 2003 states that an application within a CIZ would normally be refused, or subject to limitations, unless the applicant can demonstrate there will be no cumulative impact on the licensing objectives.[13] A common perception therefore follows that CIZs limit alcohol availability in areas already saturated. Previous work undertaken in the London Borough of Southwark found no evidence that CIZ establishment reduced the number of licence applications or issued licences across its three policy areas combined,

when compared with a control region. On analysing one CIZ area in isolation, establishment was associated with a statistically significant 133% increase in successful applications.[12]

Qualitative research, conducted in 2016, investigating the drivers behind CIZ implementation has suggested them to be more nuanced than simply restricting outlet density.[16] Participants described how CIZs have evolved to encourage local regeneration goals. CIZs tended not to be established to cap the number of licensed outlets, rather they are viewed as a tool to control the temporal availability of alcohol and even to encourage a certain type of application for example, from arts, food and coffee led establishments as opposed to 'vertical drinking' bars.

Plans to diversify night-time economies and offer alternatives to alcohol-led activities are emerging. In April 2017, the Mayor of London published guidance for culture and the night-time economy in the capital and identified diversification and inclusiveness as key areas of focus. Through encouraging leisure and cultural venues, for example, late-night markets, museums, shops, cafes, theatres and fairs, urban centres hope to attract a wider clientele. Examples of which include non-drinkers, older people, families and those with a disability.[17] Despite these plans, an evidence base in support of diversifying the night-time economy is lacking.

### Gaps in the literature
A key research gap exists around the assumption that all on-licensed and off-licensed outlets are treated as equal.[9] Although aggregate studies may show an association between outlet density and harm, they are of limited use to policy makers. Interventions need to be targeted towards outlets that cause the most harm. Therefore, through stratifying by outlet type, we aim to understand how CIZ establishment is associated with the number of licence applications and issued licences for different outlet types. Qualitative evidence indicates that CIZs can facilitate the proliferation of certain types of premises, thereby diversifying local night-time economies.[16] We therefore propose a quantitative methodology to test for this.

### Southwark's three CIZs
Southwark is an inner London borough with an estimated population of 311 655.[18] Southwark is home to approximately 1400 licensed premises which make a significant contribution to its culture and economy. However, some locations within the borough have reached saturation point. Within these areas of saturation, Southwark enforces three CIZs (figure 1); Camberwell CIZ and Peckham CIZ were both established on 5 November 2008. Bankside CIZ was established a year later on 5 November 2009. The implementation of each CIZ was to achieve an unique objective (D Franklin, Personal communication, 2018). With almost 70% of all CIZ licence applications falling within Bankside alone, the intervention aimed to tackle binge drinking-related issues. The Bankside CIZ policy applies to all venue types except for theatres and

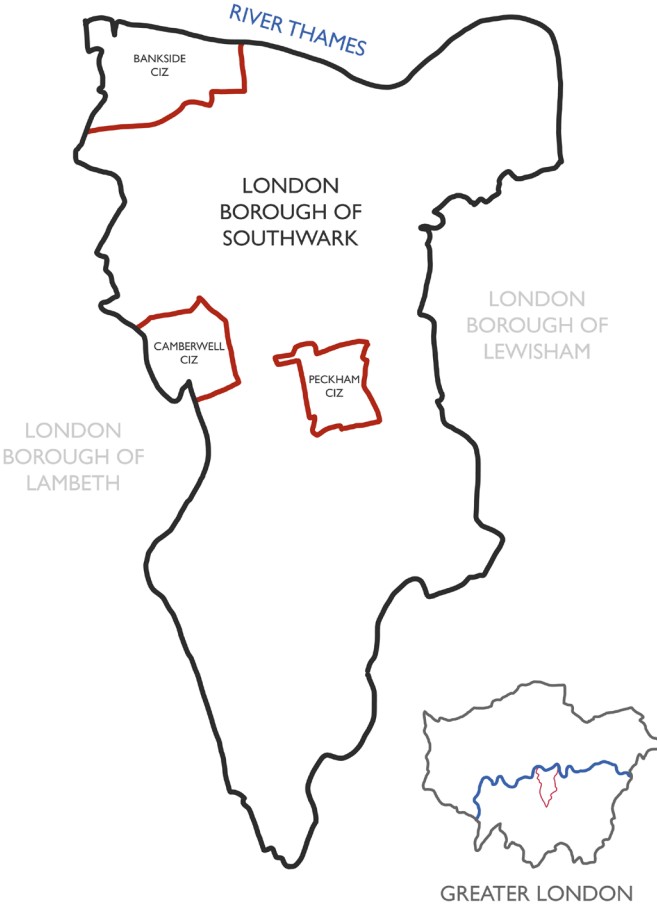

**Figure 1** London Borough of Southwark's three cumulative impact zones (inset: map of Greater London with the London Borough of Southwark highlighted).

cinemas. Peckham and Camberwell CIZs were established to tackle late-night antisocial behaviour and street drinking respectively. Notably, food-led establishments, such as restaurants and cafes, are exempt from both CIZ policies. The size and estimated population of Southwark's CIZ areas are detailed in online supplementary table 1.

## METHODS
### Data collection and collation
A comprehensive alcohol licensing data set of all applications made within Southwark between 1 April 2006 and 31 March 2017 was obtained from Authority Public Protection (APP); a database used by the council to manage licence applications.[19] Licensing data prior to 2006 could not be analysed by year of application and year of issue, as in 2005 Southwark transitioned their licensing data to APP from a previous data base. Geographical information software[20] was used to assign each outlet application an easting and northing. These coordinates were then used to identify applications within each CIZ region and the control group. The issue date was used as a proxy that applications were successful. All applications without an issue date were categorised as having been rejected or

withdrawn. An outlet type field was obtained from APP containing 100 different entries for outlet type. This field was aggregated into five mutually-exclusive categories:

▶ Bars, public-houses, hotel bars and nightclubs were grouped and are collectively referred to as *drinking establishments.*
▶ Restaurants and cafes were aggregated together and are collectively referred to as *eateries.*
▶ Licensed takeaway food restaurants were kept separate and are referred to as *takeaways.*
▶ Off-licences, grocery stores and supermarkets are grouped into one *off-sales* category.
▶ Those alcohol-outlets that do not fit into any of the above categories are collectively referred to as *other outlets* (such as florists, boats and schools).

To ensure reproducibility of the results, two researchers completed the categorisation process independently, before a final list was agreed.

Licence applications were grouped by financial year (April to the following March) starting from 2006/07. An indicator variable for the intervention was fitted, to indicate which time periods in which regions had a CIZ in operation. The three CIZs were established on 5 November in each of the respective years. We therefore grouped applications in 365 day periods relative to the date of CIZ establishment.

### Statistical analyses
Poisson regression[21] was employed to model the association between CIZ establishment (indicator variable) and first the number of applications and second the number of licences issued. The assumptions of the Poisson model were checked and the data were consistent with equidispersion (the mean and variance being equal). We adjusted for financial yearly trend, across all regions tested, and for geographical area, including both in the model as categorical adjustment variables; financial yearly trend did not fit well enough to a linear model for that to be a viable alternative. Therefore, the resulting incidence rate ratios (IRRs) for CIZs represent ratios of changes on introduction of a CIZ into an area, relative to the changes that occur with time in all regions (in the absence of any change in CIZ status and including in control regions). An adjustment was made to accommodate for the CIZs being established mid-financial year. Statistical analyses were executed using STATA V.13.0.[22] IRRs were calculated along with 95% CIs and p-values

### Patient and public involvement
There was no patient or public involvement in this study.

## RESULTS
### Descriptive findings
A full breakdown of the number of licence applications, and percentage issued, in each CIZ and the control group by financial year is provided in online supplementary table 2.

**Table 1** Number of licence applications made within each cumulative impact zone (CIZ) and the control group, by outlet type with percentages indicating, for each licence type, the proportion in the CIZ areas or the control area, 2006/07–2015/16

| | Drinking establishments | | Eateries | | Takeaways | | Off-sales | | Other outlets | | Total | |
|---|---|---|---|---|---|---|---|---|---|---|---|---|
| | n | % | n | % | n | % | n | % | n | % | n | % |
| All CIZs | 89 | 37.6 | 172 | 40.3 | 32 | 36.8 | 68 | 27.4 | 120 | 47.1 | 481 | 38.4 |
| Bankside | 63 | | 114 | | 18 | | 39 | | 97 | | 331 | |
| Peckham | 16 | | 34 | | 9 | | 18 | | 18 | | 95 | |
| Camberwell | 10 | | 24 | | 5 | | 11 | | 5 | | 55 | |
| Control | 148 | 62.4 | 255 | 59.7 | 55 | 63.2 | 180 | 72.6 | 135 | 52.9 | 773 | 61.6 |
| Total (%) | 237 (18.9) | | 427 (34.1) | | 87 (6.9) | | 248 (19.8) | | 255 (20.3) | | 1254 (100) | |

From 1 April 2006 to 31 March 2017, 1254 licence applications were received by Southwark Council (table 1). The most common application type was for eateries, with 427 (34.1%) applications made over the 10 years. The number of applications for drinking establishments, off-sales and other outlets were similar; 237 (18.9%), 248 (19.8%) and 255 (20.3%) respectively. During this time period, the council received only 87 (6.9%) applications for takeaways.

Across the period and across types of venue, 38.4% of applications were made within CIZ areas. 'Other outlets' (venue type) had the highest proportion of applications within CIZs (47.1%), of which over 80% were made within Bankside CIZ alone. 'Off-sales' venue type had the lowest proportion of applications within CIZ areas with 72.6% relating to the control region. Over a quarter (26.4%) of all new applications were made in Bankside. Peckham and Camberwell consistently received fewer applications than Bankside across the period, and for all outlet categories.

Between 2009/10 and 2011/12, following CIZ establishment, the number of issued licences for drinking establishments, eateries and other outlets decreases (figure 2).

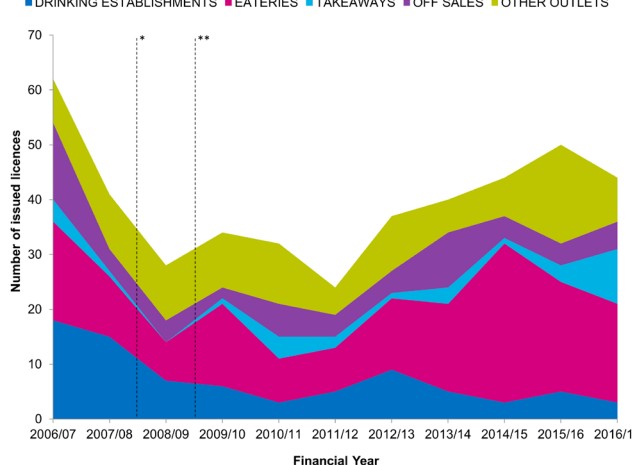

**Figure 2** Number of issued licences per financial year in cumulative impact zone (CIZ) areas, by outlet type. *Date of CIZ establishment, Peckham and Camberwell. **Date of CIZ establishment, Bankside.

Numbers for off-sales and takeaways remained relatively stable. Although the number of new drinking establishments remained low, there was a general trend upwards in the number of new eateries and 'other outlets' between 2011/12 and 2016/17.

Following CIZ implementation, the mean number of applications and issued licences for drinking establishments decreased across the three intervention areas (table 2). The reductions were not uniform and ranged from a 36% reduction in Peckham to a 100% reduction in Camberwell. Applications and issued licences for eateries and takeaways increased across the intervention regions. Notably, Peckham experienced a sevenfold increase in the mean number of newly issued licences for eateries and Camberwell experienced a 50% increase. For off-sales, there was a decrease in the number of new licences issued in both Bankside and Peckham, with Camberwell experiencing a 163% increase. Other outlets increased by 71% in Peckham, with the other two intervention areas seeing a decrease in both the mean number of applications and issued licences.

### Regression analysis

Multivariate regression analysis indicated that there was no discernible association between CIZ establishment (adjusted for overall time trends across all regions) and the number of applications and issued licences for drinking establishments across all CIZs and for each CIZ region individually (table 2). For eateries, there is some evidence that CIZ implementation was associated with a 58% increase in the number of licences issued (IRR=1.58, 95% CI: 1.02 to 2.45, p=0.04). This association is driven by a 937% increase in the number of new eateries in Peckham (IRR=10.38, 95% CI: 1.39 to 77.66, p=0.02). There was some evidence for an increase in the number of applications and issued licences for takeaways across the three intervention regions, adjusted for overall time trends across all regions (IRR=3.57, 95% CI: 1.23 to 10.24, p=0.02) and (IRR=3.89, 95% CI: 1.32 to 11.49, p=0.01) respectively. On disaggregating the areas, there was only appreciable evidence for the number of newly issued takeaways in Bankside (IRR=4.38, 95% CI: 1.20 to 15.91, p=0.03). For off-sales and other outlets, there

**Table 2** Before and after analysis of Southwark's three CIZs by outlet type, 2006/2007–2015/2016. IRRs in CIZ region(s) relative to changes over time, allowing for region

Licence applications and issued licences by venue type

| | Pre-CIZ Mean per financial year | Post-CIZ Mean per financial year | Change | Adjusted model* IRR (95% CI) | P value |
|---|---|---|---|---|---|
| **DRINKING ESTABLISHMENTS** | | | | | |
| *Number of applications* | | | | | |
| All CIZs | 11.5 | 4.7 | −59% | 1.433 (0.837 to 2.455) | 0.19 |
| Bankside | 7.3 | 4.3 | −41% | 1.340 (0.745 to 2.411) | 0.33 |
| Peckham | 2.3 | 1.5 | −36% | 1.704 (0.601 to 4.834) | 0.32 |
| Camberwell | 5.0 | 0 | −100% | – | – |
| Control | 13.5 | | | | |
| *Number of issued licences* | | | | | |
| All CIZs | 10.3 | 4.3 | −58% | 1.344 (0.763 to 2.368) | 0.31 |
| Bankside | 7.0 | 3.8 | −46% | 1.253 (0.675 to 2.328) | 0.47 |
| Peckham | 2.3 | 1.3 | −43% | 1.556 (0.530 to 4.569) | 0.42 |
| Camberwell | 3.0 | 0 | −100% | – | – |
| Control | 8.1 | | | | |
| **EATERIES** | | | | | |
| *Number of applications* | | | | | |
| All CIZs | 10.8 | 12.6 | 17% | 1.484 (0.972 to 2.266) | 0.07 |
| Bankside | 8.8 | 9.9 | 13% | 1.223 (0.767 to 1.951) | 0.40 |
| Peckham | 1.0 | 3.6 | 256% | 5.083 (1.185 to 21.803) | 0.03 |
| Camberwell | 2.0 | 3.0 | 50% | 1.869 (0.712 to 4.905) | 0.20 |
| Control | 23.2 | | | | |
| *Number of issued licences* | | | | | |
| All CIZs | 10.0 | 12.1 | 21% | 1.582 (1.020 to 2.452) | 0.04 |
| Bankside | 8.3 | 9.3 | 12% | 1.260 (0.777 to 2.041) | 0.35 |
| Peckham | 0.5 | 3.4 | 589% | 10.377 (1.386 to 77.664) | 0.02 |
| Camberwell | 2.0 | 3.0 | 50% | 1.916 (0.727 to 5.043) | 0.18 |
| Control | 21.1 | | | | |
| **TAKEAWAYS** | | | | | |
| *Number of applications* | | | | | |
| All CIZs | 2.0 | 2.1 | 7% | 3.571 (1.225 to 10.242) | 0.02 |
| Bankside | 2.0 | 2.8 | 40% | 3.500 (0.981 to 12.492) | 0.05 |
| Peckham | 1.0 | 1.6 | 60% | 2.400 (0.248 to 23.236) | 0.45 |
| Camberwell | 1.0 | 1.3 | 33% | 6.330 (0.660 to 63.639) | 0.12 |
| Control | 5.0 | | | | |
| *Number of issued licences* | | | | | |
| All CIZs | 2.0 | 2.0 | 0% | 3.894 (1.320 to 11.492) | 0.01 |
| Bankside | 2.0 | 2.8 | 40% | 4.375 (1.203 to 15.911) | 0.03 |
| Peckham | 1.0 | 1.2 | 20% | 1.667 (0.161 to 17.257) | 0.67 |
| Camberwell | 1.0 | 1.3 | 33% | 7.600 (0.746 to 77.431) | 0.09 |
| Control | 4.5 | | | | |
| **OFF-SALES** | | | | | |
| *Number of applications* | | | | | |
| All CIZs | 8.3 | 5.4 | −36% | 1.393 (0.773 to 2.510) | 0.27 |
| Bankside | 5.0 | 3.0 | −40% | 1.092 (0.536 to 2.221) | 0.81 |

**Table 2** Continued

Licence applications and issued licences by venue type

| | Pre-CIZ Mean per financial year | Post-CIZ Mean per financial year | Change | Adjusted model* | |
| --- | --- | --- | --- | --- | --- |
| | | | | IRR (95% CI) | P value |
| Peckham | 3.0 | 1.7 | −43% | 1.275 (0.453 to 3.584) | 0.64 |
| Camberwell | 1.3 | 3.5 | 163% | 3.650 (1.002 to 13.298) | 0.05 |
| Control | 16.4 | | | | |
| *Number of issued licences* | | | | | |
| All CIZs | 7.3 | 4.9 | −34% | 1.535 (0.824 to 2.861) | 0.18 |
| Bankside | 4.3 | 3.0 | −31% | 1.316 (0.625 to 2.769) | 0.47 |
| Peckham | 2.5 | 1.1 | −54% | 1.067 (0.331 to 3.434) | 0.91 |
| Camberwell | 1.3 | 3.5 | 163% | 3.350 (0.918 to 12.223) | 0.07 |
| Control | 14.6 | | | | |
| **OTHER OUTLETS** | | | | | |
| *Number of applications* | | | | | |
| All CIZs | 10.0 | 6.9 | −31% | 0.623 (0.371 to 1.044) | 0.07 |
| Bankside | 9.3 | 7.5 | −19% | 0.556 (0.325 to 0.951) | 0.03 |
| Peckham | 1.0 | 2.3 | 129% | 2.171 (0.466 to 10.121) | 0.32 |
| Camberwell | 1.0 | 1.0 | 0% | 1.143 (0.120 to 10.876) | 0.91 |
| Control | 12.3 | | | | |
| *Number of issued licences* | | | | | |
| All CIZs | 9.0 | 5.9 | −35% | 0.601 (0.346 to 1.047) | 0.07 |
| Bankside | 8.3 | 6.5 | −21% | 0.571 (0.321 to 1.013) | 0.55 |
| Peckham | 1.0 | 1.7 | 71% | 1.643 (0.339 to 7.962) | 0.54 |
| Camberwell | 1.0 | 0.8 | −25% | 1.041 (0.101 to 10.692) | 0.97 |
| Control | 10.5 | | | | |

No intervention was implemented in the control area and therefore a mean across all financial years is displayed.
– too few applications were made to perform regression analysis.
*Licence applications received and (in other models) issued according to area, reported by outlet type, analysed using Poisson regression. These models are all adjusted for financial year, and for geographical area, including both in the model as categorical adjustment variables. The number of applications and issued licences preintervention was used as the reference group (IRR=1.00).
CIZ, cumulative impact zones; IRR, incidence rate ratio.

was little evidence for associations between CIZ establishment and the number of applications and issued licences across all CIZs. Camberwell CIZ experienced a substantial relative increase in the number of off-sale applications (IRR=3.65, 95% CI: 1.00 to 13.30, p=0.05), but with only weak evidence for this association. Bankside CIZ shows some evidence for a decrease in the number of applications for other outlets (IRR=0.56, 95% CI: 0.33 to 0.95), p=0.03; however other regions reported relative decreases, although with no ability to rule out random chance as an explanation.

## DISCUSSION
### Statement of principal findings
Across the three intervention areas, absolute numbers of issued licences for drinking establishments decreased and remained low in frequency following CIZ establishment. In Camberwell no single application for a new drinking establishment was made postintervention. Conversely,

the number of new eateries increased year-on-year. The average number of issued licences for drinking establishments, off-sales and other outlets decreased following CIZ establishment, except for off-sales in Camberwell and other outlets in Peckham. The mean number of new eateries and new takeaways increased in all areas tested. Regression analyses of outlet type and intervention gave some evidence for a positive association for new eateries across all CIZs and for new eateries in Peckham CIZ, compared with the control area. For takeaways there was some evidence for an increase in the number of issued licences both across all CIZs and for Bankside CIZ, compared with the control area.

### Strengths and weaknesses in relation to other studies and policy implications
The findings from this study support those from previous research that CIZs can be used as a tool to shape local alcohol environments,[16] yet this is the first quantitative evidence to substantiate the theories. This is best

exemplified by Peckham CIZ which was established to tackle late-night antisocial behaviour occurring due to increasing numbers of drinking establishments in the local area. Food-led establishments were exempted from the CIZ policy in Peckham. We found some evidence the area experienced a substantial increase in licensed eateries following CIZ establishment.

SOLP are an opportunity for LAs to set the licensing goals of urban areas *a priori*. Excluding eateries from the CIZ policy in Peckham was successful in drawing new food-led establishments to that area. Although not explicitly stated in Southwark's SOLP, Camberwell CIZ was established to tackle issues relating to street drinking. Yet the number of new off-licences in Camberwell appeared to increase following CIZ establishment and there were no new applications for drinking establishments in Camberwell postintervention. In Bankside CIZ, a reduction in new Drinking Establishments was experienced. Although, this is in line with the goals of the Bankside policy, an increase in new licensed takeaways is unlikely to be. These findings imply that descriptive analysis of the absolute numbers of newly licensed premises in a particular geographic location is inadequate. A robust statistical test, which compares intervention areas to a control region, is required to fully appreciate the impact of CIZ policy. The results from this study can be used to inform the renewal of Southwark's SOLP and provide an evidence-base from which licensing goals to diversify the night-time economy can be developed.

### Strengths, weaknesses and further work

Stratifying the dataset by financial year, geographic area and outlet type rendered some strata with very few, or in some cases zero, counts. This was particularly an issue for Peckham CIZ and Camberwell CIZ, which accounted for only 7.6% and 4.4% of the data respectively. Statistical power was therefore constrained resulting in an increased risk of type 2 error.

Aggregation of outlet type into five discrete categories was a subjective task. Although the approach taken was as systematic as possible, and a sample of venues was sense checked against their online descriptions, there remains the possibility for interpretation. For example, an outlet which operates as a cafe in the daytime, and a bar at night could be categorised as either 'drinking establishment', 'eatery' or even 'other outlet'. To ensure reproducibility of the results, two researchers completed the categorisation process independently, before a final list was agreed.

Limiting outlet type to only five categories was restrictive and may have reduced precision, but necessary to provide suitable observations in each category. Categorising small drinking establishments that, for example offer food, provide seating areas for their customers and close at 23:00, as equal to a 1000-person capacity nightclub with vertical drinking until 06:00 is potentially problematic. Policy makers may unnecessarily restrict the former type of venue as a consequence. To better identify the types of venues that cause the most harm, further

work should consider other methods of stratifying outlet type, for example by opening time and venue capacity.

A number of gaps exist in the literature preventing policy makers from making informed recommendations relating to alcohol-availability interventions. Evidence for alcohol-availability polices are from aggregate studies. Given the heterogeneous nature of the outlet types within on-licence and off-licence categories, studies providing a disaggregated analysis are needed. This study provides a quantitative evaluation of CIZs within a major metropolitan area that disaggregates outlet type into five categories. This novel approach exemplifies how local policy can be used to support certain licensing goals and can facilitate the diversification of the night-time economy.

This research provides deeper insight as to the nature of alcohol exposure in different locations within an urban centre. Further work is required to understand which outlet types are associated with the greatest amount of alcohol-related harm.

Southwark is an inner London Borough with a thriving night-time economy. We argue these findings from this research are generalisable to similar urban centres both in the UK and internationally. The methodology employed is reproducible and scalable. Further research should attempt to replicate similar results in other LAs. In order to achieve higher statistical power, similar studies across a cluster of LA, or on a regional or national scale are recommended.

### CONCLUSIONS

CIZs can be used as policy levers to shape local alcohol environments to support the licensing goals of specific geographical areas and diversify the night-time economy. While we found no evidence that CIZs limit the density of different types of licensed venues, CIZs may be used as a means to encourage certain types of outlet over others and thereby change the tone of an area's night-time economy. To further support the development of local licensing policies, an evidence-base for the diversification of the night-time economy is needed, and further work to understand the impact of different outlet types on alcohol-related harm outcomes is required.

**Acknowledgements** The authors would like to acknowledge Southwark Council's Licensing Authority for providing access to the data for this study and for providing valuable context surrounding the local licensing policy.

**Contributors** CAS completed the literature review for this study, led on the data analysis and drafted the manuscript. DF provided context as to the local licensing policy. HW supported the development of the statistical model. CW created figure 1 and sourced the data for supplementary table 1. RJP critically reviewed the manuscript and revised a second draft. HW, AP and RJP inputted into the data interpretation and revised the final manuscript. All authors have approved the final manuscript.

**Funding** This work was supported by the National Institute for Health Research (NIHR) under the Collaborations for Leadership in Applied Health Research and Care (CLAHRC) programme for North West London.

**Disclaimer** The views expressed in this publication are those of the author(s) and not necessarily those of the NHS, the NIHR or the Department of Health.

**Map disclaimer** The depiction of boundaries on the map(s) in this article do not imply the expression of any opinion whatsoever on the part of BMJ (or any member of its group) concerning the legal status of any country, territory, jurisdiction or area or of its authorities. The map(s) are provided without any warranty of any kind, either express or implied.

**Competing interests** None declared.

**Patient consent for publication** Not required.

**Provenance and peer review** Not commissioned; externally peer reviewed.

**Data availability statement** No data are available.

**Open access** This is an open access article distributed in accordance with the Creative Commons Attribution 4.0 Unported (CC BY 4.0) license, which permits others to copy, redistribute, remix, transform and build upon this work for any purpose, provided the original work is properly cited, a link to the licence is given, and indication of whether changes were made. See: https://creativecommons.org/licenses/by/4.0/.

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
