## [Reviewer comments · BMJ Open]

ARTICLE DETAILS

TITLE (PROVISIONAL)	An observational study to examine how Cumulative Impact Zones influence alcohol availability from different types of licensed outlets in an inner London Borough
AUTHORS	Sharpe, Carolyn; Poots, Alan; Watt, Hilary; Williamson, Chris; Franklin, David; Pinder, Richard

VERSION 1 – REVIEW

REVIEWER	Kerri Coomber Deakin University, Australia
REVIEW RETURNED	12-Nov-2018

GENERAL COMMENTS	Thank you for the opportunity to review this paper. This study examines the impact of Cumulative Impact Zones on the number of new liquor licenses issued within the Borough of Southwark. 1. Some context around why the Borough of Southwark was chosen (and not other areas in London) is needed.2. It would be good to have more details about the Borough of Southwark and each CIZ. For instance, population, km² area, type of nightlife etc, to get a sense of how representative this area is of the rest of London (or England and Wales).3. Further detail around the coding of the license types is needed. Please provide some clear definitions of each type, including example types of outlets. This will help the reader differentiate between the types (for instance, it may not be apparent what the difference between take-aways and off-license are).4. Further to this, you state that some outlets could be classified into more than one category of license. What criteria were used to code the outlet into one category as opposed to another; that is, for example, what was deemed the more important feature of the outlet that resulted in being coded as a drinking establishment rather than an eatery?5. Information on the control site is needed. For instance, the location, population, and so forth. Ideally, this control site would be as similar to Southwark as possible, but just without CIZs.6. You report on applications and issued licenses only, do you have data on cancelled licenses? This would provide important contextual information on the absolute number of outlets in each CIZ and the control area. By examining real growth (that is, number issued – number cancelled for each outlet type) you
---

	would get a better sense of whether outlet density is increasing for each outlet type. 7. The point above, and the issue of lack of information about the areas under study, become important when you are reporting figures such as a 589% increase in issued eatery licenses in Peckham (Table 2). When looking at absolute numbers (M=0.5 to M=3.4), this may not be that large of an increase if, say, an average of 1 eatery license per year was cancelled and Peckham was a large area with a large population. Some more context is needed to more adequately interpret these changes. 8. It would be good to see some discussion around how changing the types of licensed outlets within particular areas may then influence rates of harms (e.g., assaults, under-age drinking, ambulance attendances). You specifically state that the goal of the Peckham CIZ was to reduce late-night anti-social behaviour, and while there has been an increase in eateries, no evidence is presented that this change in outlet type reduced anti-social behaviour in this area. Similarly, it is stated that the Camberwell CIZ goal was to reduce on-street drinking; while the number of off-licenses issues increased, we do not actually know if this resulted in a corresponding increase in drinking in the street.
--	--

REVIEWER	Raimee Eck Fellow, Division of Cancer Control and Population Sciences Behavioral Research Program, National Cancer Institute, United States
REVIEW RETURNED	12-Feb-2019

GENERAL COMMENTS	Thank you for the opportunity to review this manuscript. I hope my comments are found to be helpful. This manuscript could be an interesting contribution to the literature around alcohol outlets and policy, but a number of issues should be addressed by the authors. Overall the intro could clarify a little more about what the CIZs are and what the expected findings would be if the CIZs were operating correctly. And a little more detail is needed in the methods. Intro: Instead of Box 1 with the licensing objectives, which could easily go in the text, consider replacing with the rationale, goals, and parameters for establishment of the CIZs, as that is unclear. Also, what is the coverage area of these CIZs? Is there an average size (e.g., 2 city blocks or an entire neighborhood)? A little more background would help set the reader up. Finally, you mention the diversification of night-time economies—is this a goal of the CIZs and is there public health-related evidence behind it? If so, it should be stated and referenced. And briefly, I believe the last paragraph could go in the methods section (describes the CIZs). Methods: Indicate why you chose these years for the licenses. Perhaps include a map of Southwark and identify the three CIZs and control area in addition to some basic descriptors of each CIZ and the control area (population, avg income, area, etc). Also, how was the control area chosen? Were confounders across the CIZs and control area not considered necessary for the regressions? I think a rationale of why you only chose to control for what you did is necessary. I recommend a description (or table) of the five categories (e.g., I'm not sure what a takeaway is, are cafes and
---

	restaurants included in “eateries” as they are excluded from the CIZ policies, and do “eateries” serve alcohol, too). I’d also consider breaking down the “other” category a bit more, especially since that is the highest proportion of applications within the CIZs and one of the goals of the analysis is to identify which types of outlets are impacted by the implementation of CIZs. With “other,” you have no idea what about half the licenses are. Results: In the results you report descriptive comparisons between the number of applications in the CIZs compared to each other and to the control area, which I would argue is not very meaningful if we don’t know the size or composition of each area (i.e., what is the denominator?). Discussion: Succinct statement of findings. I appreciated the discussion of limitations and needs for policymakers. Please elaborate more on the number or zero cells in the data; are there any other methods you could have considered to deal with those (perhaps this goes in the methods section instead)? The first three sentences of the Strengths and weaknesses section belongs in the intro to help set the stage. This was very helpful to understand the setting. You also say on page 14 line 5, “In Bankside CIZ, a non-significant reduction in Drinking Establishments was experienced in the area.” But the finding is not a reduction in in establishments; it was a reduction in the number of applications and the number of licenses issued (unless I have an incorrect understanding of this). I recommend going through the entire manuscript to ensure this language is accurate. And again, since you indicate in your conclusions that this research shows that diversification in the night time economy is demonstrated here, I suggest including an evidence base for this type of policy outcome.
--	--

VERSION 1 – AUTHOR RESPONSE

Reviewer 1

1. Some context around why the Borough of Southwark was chosen (and not other areas in London) is needed.

Response: The London Borough of Southwark was chosen as three of the five authors were either working at or affiliated with the borough at the time of undertaking the research. Although licensing data is a valuable resource, it is not publicly available. Moreover, different London boroughs have different methods and systems for capturing licensing data, many of which do not permit interrogation at the level of rigor which was performed for this analysis. Therefore, Southwark was selected as high quality data were collected which was readily available.

2. It would be good to have more details about the Borough of Southwark and each CIZ. For instance, population, km² area, type of nightlife etc, to get a sense of how representative this area is of the rest of London (or England and Wales).

Response: A map of Southwark with its three Cumulative Impact Zones highlighted has been included (FIG 1) Additionally, supplementary table 1 provides information on the size of each CIZ area (and the control region) as well as the estimated population size.

3. Further detail around the coding of the license types is needed. Please provide some clear definitions of each type, including example types of outlets. This will help the reader differentiate between the types (for instance, it may not be apparent what the difference between take-aways and off-license are).

Response: Further detail as to the outlet type categories has been included in the methods section (Page 7) as well as some examples of the venues included in each category.

4. Further to this, you state that some outlets could be classified into more than one category of license. What criteria were used to code the outlet into one category as opposed to another; that is, for example, what was deemed the more important feature of the outlet that resulted in being coded as a drinking establishment rather than an eatery?

Response: The authors thank the reviewer for this comment. Further detail as to the outlet type categories has been included in the methods section, as well as some examples of the venues included in each category.

The final categorisation assigned to each outlet was a judgement: we took a number of steps to ensure the categorisation process was as rigorous as possible. We acknowledge that the categorisation of outlets was a qualitative process and therefore a risk of subjectivity remains. We did our best to use the local knowledge of Southwark's licensed venues available - The Southwark Head of Licensing is an author on this paper and is very familiar with a large number of Southwark's venues. We also checked a sample of the venues against their online description. Additionally, two researchers independently categorised each venue independently. Any discrepancies were addressed by looking at online descriptions of the venue. Clarification is provided in the manuscript. A limitation has been included to acknowledge the above: "Aggregation of outlet type into five discrete categories was a subjective task. Although the approach taken was as systematic as possible, and a sample of venues was sense checked against their online descriptions, there remains the possibility for interpretation. For example: an outlet which operates as a café in the daytime, and a bar at night could be categorised as either 'drinking establishment', 'eatery' or even 'other outlet'. To ensure reproducibility of the results, two researchers completed the categorisation process independently, before a final list was agreed."

5. Information on the control site is needed. For instance, the location, population, and so forth. Ideally, this control site would be as similar to Southwark as possible, but just without CIZs.

Response: The authors thank the reviewer for their comment. A map of Southwark with its three Cumulative Impact Zones highlighted has been included (FIG 1). Additionally, supplementary table 1 provides information on the size of each CIZ area (and the control region) as well as the estimated population size.

6. You report on applications and issued licenses only, do you have data on cancelled licenses? This would provide important contextual information on the absolute number of outlets in each CIZ and the control area. By examining real growth (that is, number issued – number cancelled for each outlet type) you would get a better sense of whether outlet density is increasing for each outlet type.

Response: The analysis does not examine the changing number of licensed venues in each CIZ in Southwark, rather it evaluates new applications and newly issued licenses: this is a methodological limitation. Nevertheless, the data source does not facilitate measurement of attrition of outlets robustly - there would be too much uncertainty caused by assumptions.

7. The point above, and the issue of lack of information about the areas under study, become important when you are reporting figures such as a 589% increase in issued eatery licenses in Peckham (Table 2). When looking at absolute numbers ($M=0.5$ to $M=3.4$), this may not be that large of an increase if, say, an average of 1 eatery license per year was cancelled and Peckham was a large area with a large population. Some more context is needed to more adequately interpret these changes.

Response: As above, the analysis does not examine the changing number of licensed venues in each CIZ in Southwark, rather it evaluates new applications and newly issued licenses: this is a methodological limitation. Nevertheless, the data source does not facilitate measurement of attrition of outlets robustly - there would be too much uncertainty caused by assumptions.

8. It would be good to see some discussion around how changing the types of licensed outlets within particular areas may then influence rates of harms (e.g., assaults, under-age drinking, ambulance attendances). You specifically state that the goal of the Peckham CIZ was to reduce late-night anti-social behaviour, and while there has been an increase in eateries, no evidence is presented that this change in outlet type reduced anti-social behaviour in this area. Similarly, it is stated that the Camberwell CIZ goal was to reduce on-street drinking; while the number of off-licenses issues increased, we do not actually know if this resulted in a corresponding increase in drinking in the street.

Response: We agree that a natural and useful extension of this research would be to evaluate the effect of CIZ implementation on alcohol related outcomes including violence, health harms and anti-social behaviour. This is out of the scope of this piece of research however we have stated in our conclusion that: "further work is required to understand the impact of different outlet types on alcohol-related harm outcomes". A follow up study - that evaluates the impact of different types of licensed premises in Southwark on alcohol-related violence has been completed by the same authors and is awaiting publication.

Reviewer 2

1. Instead of Box 1 with the licensing objectives, which could easily go in the text, consider replacing with the rationale, goals, and parameters for establishment of the CIZs, as that is unclear.

Response: Box 1 has now been removed and the Licensing Objectives included in the main text. Additional context as to the rationale, goals and parameters for establishing a CIZ has been added to the introduction. The rationale and goals of the three CIZ's established in Southwark is included in the introductory text under 'Southwark's three cumulative impact zones'

2. Also, what is the coverage area of these CIZs? Is there an average size (e.g., 2 city blocks or an entire neighborhood)? A little more background would help set the reader up.

Response: A map of Southwark with it's three Cumulative Impact Zones (FIG 1) highlighted has been included. Additionally, supplementary table 1 provides information on the size of each CIZ area (and the control region) as well as the estimated population size.

3. Finally, you mention the diversification of night-time economies—is this a goal of the CIZs and is there public health-related evidence behind it? If so, it should be stated and referenced.

Response: We have stated in the introduction that plans to diversify the night-time economy in London - led by the London Mayor - are emerging. A reference for this statement has been provided. However, the evidence base for diversifying the night-time economy is minimal. We have therefore included a clarification in both the introduction and the conclusion to clarify this point.

4. And briefly, I believe the last paragraph could go in the methods section (describes the CIZs).

Response: The last paragraph describes the three CIZ policies already established in the London Borough of Southwark. This research only evaluated the impact of these policies. No changes to the policies were made during the course of the research. Hopefully this clarification is helpful, but if not we ask if the reviewer to provide more information as to why this paragraph should be included in the methods section.

5. Indicate why you chose these years for the licenses.

Response: Further clarification has been provided in the methods section: "Licensing data from prior to 2006 could not be analysed by year of application and issue, as in 2005 Southwark transitioned their licensing data to APP from a previous data base. All licences that were issued prior to the transition were assigned an issue date within 2005."

6. Perhaps include a map of Southwark and identify the three CIZs and control area in addition to some basic descriptors of each CIZ and the control area (population, avg income, area, etc). Also, how was the control area chosen?

Response: A map of Southwark with its three Cumulative Impact Zones (FIG 1) highlighted has been included. Additionally, supplementary table 1 provides information on the size of each CIZ area (and the control region) as well as the estimated population size.

7. Were confounders across the CIZs and control area not considered necessary for the regressions? I think a rationale of why you only chose to control for what you did is necessary.

Response: The adjustment was limited by the availability of robust adjustment variables. CIZ boundaries are not regular geographies. Therefore their boundaries do not align to UK output areas (OAs). OAs are part of the UK geographic hierarchy for reporting statistics for small areas. Therefore, the population of a CIZ and their area size can only be estimated. We have included an explanation as to the above as a footnote beneath Supplementary Table 1.

Therefore, confounding factors such as area size or area population could not be included. To account for this, we adjusted for the number of alcohol outlets in each area. Arguably, this is a better adjustment variable - at least for on-licensed venues - as restaurants, bars and nightclubs in London are independent of the number of people living in a particular area. These types of venues are more likely to be located where there is a high footfall of tourists, workers as well as residents. To take an example, Bankside CIZ will have a small residential population, but is the most densely populated with alcohol-outlets as this area has a high footfall of tourists and workers.

We also adjusted for the number of license application / issued licenses each financial year. This will seek to mitigate secular trends in the number of new alcohol-venues in each area (for example following the economic downturn in 2007).

8. I recommend a description (or table) of the five categories (e.g., I'm not sure what a takeaway is, are cafes and restaurants included in "eateries" as they are excluded from the CIZ policies, and do "eateries" serve alcohol, too).

Response: Further detail as to the outlet type categories has been included in the methods section, as well as some examples of the venues included in each category. To clarify, all outlets analysed had applied to the Southwark Licensing Authority for a licence to sell alcohol. Therefore, yes, eateries would service alcohol.

9. I'd also consider breaking down the "other" category a bit more, especially since that is the highest proportion of applications within the CIZs and one of the goals of the analysis is to identify which types of outlets are impacted by the implementation of CIZs. With "other," you have no idea what about half the licenses are.

Response: The authors thank the reviewer for their comment. We considered breaking down the other category further, however we decided against doing so for the following reasons: Firstly, the number of outlet type categories is a trade off with statistical power. We felt further stratification would limit the statistical power of the study. We have included a limitation that relates to this point in the discussion: "Stratifying the data set by financial year, geographic area and outlet type rendered some strata with very few, or in some cases zero, counts. This was particularly an issue for Peckham CIZ and Camberwell CIZ, which accounted for only 7.6% and 4.4% of the data respectively. Statistical power was therefore constrained resulting in an increased risk of type two error."

Secondly, we had concerns regarding the accessibility of the results to readers. We are already quoting results for licence applications and issued licences for five outlet type categories. We were concerned that the paper would become too confusing for readers to follow if further findings were reported.

10. Results: In the results you report descriptive comparisons between the number of applications in the CIZs compared to each other and to the control area, which I would argue is not very meaningful if we don't know the size or composition of each area (i.e., what is the denominator?).

Response: A map of Southwark with its three Cumulative Impact Zones (FIG 1) highlighted has been included. Additionally, supplementary table 1 provides information on the size of each CIZ area (and the control region) as well as the estimated population size.

11. Discussion: Succinct statement of findings. I appreciated the discussion of limitations and needs for policymakers. Please elaborate more on the number or zero cells in the data; are there any other methods you could have considered to deal with those (perhaps this goes in the methods section instead)?

Response: Supplementary Table 1 shows the number of licence applications and percentage issued in each CIZ and the control group by financial year, and by outlet type. There were no license applications made (or licenses issued) within the four areas analysed on multiple occasions over the study period. All data were included in the statistical model and therefore were accounted for in the before and after analysis which is presented in Table 2. In Camberwell, there were no applications for drinking establishments were made to Southwark Licensing Authority once the CIZ had been established. This is highlighted in the statement of principal findings. As such the before and after statistical analysis is not quoted as the IRR would equal 0.

12. The first three sentences of the Strengths and weaknesses section belongs in the intro to help set the stage. This was very helpful to understand the setting.

Response: The authors thank the reviewer for their comments. The first paragraph in the 'strengths and limitations' section has been moved to the introduction as suggested.

13. You also say on page 14 line 5, "In Bankside CIZ, a non-significant reduction in Drinking Establishments was experienced in the area." But the finding is not a reduction in in establishments; it was a reduction in the number of applications and the number of licenses issued (unless I have an incorrect understanding of this). I recommend going through the entire manuscript to ensure this language is accurate.

Response: The authors thank the reviewer for their comments. The wording has been changed to reflect this comment. A review of the whole manuscript to ensure accurate language has been used.

14. And again, since you indicate in your conclusions that this research shows that diversification in the night time economy is demonstrated here, I suggest including an evidence base for this type of policy outcome.

Response: We have stated in the introduction that plans to diversify the night-time economy in London - led by the London Mayor - are emerging. A reference for this statement has been provided. However, the evidence base for diversifying the night-time economy is minimal. We have therefore included a clarification in both the introduction and the conclusion to clarify this point.

VERSION 2 – REVIEW

REVIEWER	Kerri Coomber Deakin University, Australia
REVIEW RETURNED	26-Mar-2019

GENERAL COMMENTS	The authors have addressed all comments and concerns from the editor and reviewers, resulting in a much improved paper.
---

REVIEWER	Raimee Eck National Cancer Institute, USA
REVIEW RETURNED	09-Apr-2019

GENERAL COMMENTS	This revision is much improved. Thank you for allowing me to review. In the intro, you cite Gmel, 2016 to support the statement that grouping all outlets is a crude measure, which I agree with; however, in the conclusion, you make the statement, "...it has been recommended that aggregate studies are discontinued." using the same citation. I encourage you to read the response from Morrison et al. to the Gmel paper and consider a revision to your statement. There is more nuance behind this. In the methods, you indicate that area was included as a categorical variable--please include the categories and the justification for using area as a category vs continuous. Also, a brief description that the model was checked for necessary assumptions for a Poisson model (such as for overdispersion) should be included. In the discussion, you don't separate out the results between the applications and the issued licenses, nor do you discuss the implications of an increase in one outcome, but not the other in the same CIZ. Since you decided to model both outcomes, it would be helpful for the discussion to address both outcomes. In table 1, please include the years that are represented.
---

VERSION 2 – AUTHOR RESPONSE

Comment 1

In the intro, you cite Gmel, 2016 to support the statement that grouping all outlets is a crude measure, which I agree with; however, in the conclusion, you make the statement, "...it has been recommended that aggregate studies are discontinued." using the same citation. I encourage you to read the response from Morrison et al. to the Gmel paper and consider a revision to your statement. There is more nuance behind this.

Response:

Thank you for raising this point. We have considered the response from Morrison et al. to the Gmel paper and recognise the nuance highlighted. We have therefore added the following sentence to the introduction following our reference to Gmel et al "... and debate remains around when to use aggregated and disaggregated analyses" referencing the Morrison et al paper. Additionally, we have softened the language used in the discussion: The sentence "...it has been recommended that aggregate studies are discontinued." Has been replaced with "...studies providing a disaggregated

analysis are needed.”

Comment 2

In the methods, you indicate that area was included as a categorical variable--please include the categories and the justification for using area as a category vs continuous. Also, a brief description that the model was checked for necessary assumptions for a Poisson model (such as for over-dispersion) should be included.

Response:

We fitted region as a categorical variable consisting of three CIZ regions and one control region (using an indicator variable for each CIZ region, with the control region taken as the baseline). We also fitted a binary variable for CIZ to indicate the time periods when a CIZ was enforced in a region. Therefore the resulting incidence rate ratios (IRR's) for CIZ's represent ratios of changes on introduction of a CIZ into an area, relative to average changes over time that occur across all regions (in the absence of any change in CIZ status and including in control regions). We did not allow for the physical size of regions, since we wanted to allow for the number of licence applications received in practice (and then the number issued in practice) at baseline (in 2006/7), so that reported IRR's in later years referred to relative changes in these numbers relative to baseline). Once this is allowed for, additional allowance for physical size (which depends on region and does not change over time) would be redundant.

We reviewed the interpretation of these IRRs throughout the document. In some places we felt the interpretation could be improved, so we have now clarified accordingly.

All assumptions of the Poisson model were checked with no observed over dispersion. We have added a statement about this near the start of the “statistical analysis” methods section.

Comment 3

In the discussion, you don't separate out the results between the applications and the issued licenses, nor do you discuss the implications of an increase in one outcome, but not the other in the same CIZ. Since you decided to model both outcomes, it would be helpful for the discussion to address both outcomes.

Response:

Thank you for the comment. The difference in the results for number of license applications and number of licenses issued are not generally qualitatively different. Hence we do not feel that it is appropriate to focus on any such possible differences in the discussion. Given also the word limit constraints and the stage of the review process, we would propose retaining the current content. However, we are willing to add content if the editors deem this content should be included.

Comment 4

In table 1, please include the years that are represented.

Response:

Thank you. The table description has been updated to include the years to which the data refers.

Further changes:

We have responded to recent Nature comment “Statisticians rise up against significance testing” (with 800 signatures) and a related series of 45 articles published in *The American Statistician* (in March of this year), by removing reference to the 5% cut-off, and interpreting results in terms of strength of evidence, rather than strictly in relation to $p < 0.05$ cut-off.

Amrhein V, Greenland S, McShane B. Statisticians rise up against significance testing. *Nature* 567, 305-307 (2019). doi: 10.1038/d41586-019-00857-9.

VERSION 3 – REVIEW

REVIEWER	Raimee Eck National Cancer Institute, USA
REVIEW RETURNED	20-Aug-2019
GENERAL COMMENTS	The edits included are appreciated and increase the comprehension in several areas. No further comments. Best of luck.